# How to Learn a Useful Critic? Model-based Action-Gradient-Estimator Policy Optimization

**Pierluca D'Oro**[*]
MILA, Université de Montréal
pierluca.doro@mila.quebec

**Wojciech Jaśkowski**
NNAISENSE
wojciech@nnaisense.com

## Abstract

Deterministic-policy actor-critic algorithms for continuous control improve the actor by plugging its actions into the critic and ascending the action-value gradient, which is obtained by chaining the actor's Jacobian matrix with the gradient of the critic with respect to input actions. However, instead of gradients, the critic is, typically, only trained to accurately predict expected returns, which, on their own, are useless for policy optimization. In this paper, we propose MAGE, a model-based actor-critic algorithm, grounded in the theory of policy gradients, which explicitly learns the action-value gradient. MAGE backpropagates through the learned dynamics to compute gradient targets in temporal difference learning, leading to a critic tailored for policy improvement. On a set of MuJoCo continuous-control tasks, we demonstrate the efficiency of the algorithm in comparison to model-free and model-based state-of-the-art baselines.

## 1  Introduction

Reinforcement learning (RL) [36, 52] studies sequential decision making problems, in which an agent aims at maximizing the cumulative reward it collects in an environment. One of the most popular classes of algorithms for RL are *policy gradient methods* [10, 53], which involve differentiable control policies improved by gradient ascent. They feature suitability to environments with continuous state and action spaces, and compatibility with state-of-the-art deep learning [42] methods. Policy gradient algorithms often employ an actor-critic [26] scheme: an *actor*, which determines the control policy, is evaluated using a *critic*. Thus, the degree of actor's improvement is limited by the information provided by the critic, naturally raising the question of how the critic should be trained.

Typically, algorithms that use powerful function approximators [18, 28] learn the critic by temporal difference [50], optimizing for an accurate prediction of the expected return of the actor. For deterministic-policy continuous-control [28, 47], however, the value provided by the critic is neither used for improving the policy nor for acting in the environment [53]. Instead, only the *action-gradient* of the value function, i.e., the gradient of the critic w.r.t. the action performed by the actor, is employed during policy optimization. Specifically, the policy gradient is obtained through the computation of the *action-value gradient*, by chaining the actor's Jacobian with the action-gradient of the critic.

Learning the critic *by value* rather than *by action-gradient* of the value relies on hazy smoothness assumptions on the real value function [47]. This means that, in conventional temporal difference learning, the critic learns action-value gradients *implicitly*, which could harm the performance of a deterministic policy gradient algorithm.

In this paper, we propose Model-based Action-Gradient-Estimator Policy Optimization (MAGE), a continuos-control deterministic-policy actor-critic algorithm that *explicitly* trains the critic to provide accurate action-gradients for the use in the policy improvement step. Motivated by both the theory on

---

[*]Work done while at NNAISENSE.

*Deterministic Policy Gradients* [47] and practical considerations, MAGE uses temporal difference methods to minimize the error on the action-value gradient. For this, the algorithm leverages a trained dynamics model as a proxy for a differentiable environment and techniques reminiscent of double backpropagation [12]. On a challenging continuous control benchmark [6, 55], we show that MAGE is significantly more sample-efficient than state-of-the-art model-free and model-based baselines.

The rest of the paper is organized as follows. In Section 2, we provide the notation and background on deterministic policy gradients. Our algorithm, together with its theoretical motivation, is introduced in Section 3, followed by empirical results in Section 4. In Section 5, we present some of the related work and its relationship with our approach.

## 2 Background

### 2.1 Preliminaries

Consider a discrete-time Markov Decision Process [36] (MDP), defined as $\mathcal{M} = (\mathcal{S}, \mathcal{A}, p, r, \gamma, \mu)$, where $\mathcal{S}$ is the space of possible states, $\mathcal{A}$ is the space of possible actions, $p : \mathcal{S} \times \mathcal{A} \to \Delta(\mathcal{S})$ is the transition model, $r : \mathcal{S} \times \mathcal{A} \to \mathbb{R}$ is the known and differentiable reward function, $\gamma$ is the discount factor, $\mu \in \Delta(\mathcal{S})$ is the initial state distribution. The behavior of the agent is described by a deterministic policy $\pi_{\boldsymbol{\theta}} : \mathcal{S} \to \mathcal{A}$, belonging to a parametric space of policies $\Pi = \{\pi_{\boldsymbol{\theta}} : \boldsymbol{\theta} \in \Theta \subseteq \mathbb{R}^n\}$, for which we will occasionally omit the parameter subscript. Let $d_{\mu}^{\pi}$ be the $\gamma$-discounted state distribution induced by policy $\pi_{\boldsymbol{\theta}}$, defined as $d_{\mu}^{\pi}(s) = (1 - \gamma) \sum_{t=0}^{\infty} \gamma^t \Pr(s_t = s | \pi, \mu)$. The total reward collected by an agent is quantified with action-value function $Q^{\pi}(s, a) = \mathbb{E}\left[\sum_{t=0}^{\infty} \gamma^t r(s_t, a_t) | s_0 = s, a_0 = a\right]$ and performance function $J(\boldsymbol{\theta}) = \mathbb{E}_{s \sim \mu}[Q^{\pi}(s, \pi_{\boldsymbol{\theta}}(s))]$.

Practical algorithms can employ an approximate action-value function $\widehat{Q}$ and an approximate dynamics model $\widehat{p}$, which, most commonly, are parametric function approximators specified by the spaces $\mathcal{Q} = \{Q_{\boldsymbol{\phi}} : \boldsymbol{\phi} \in \Phi \subseteq \mathbb{R}^h\}$ and $\mathcal{P} = \{p_{\boldsymbol{\omega}} : \boldsymbol{\omega} \in \Omega \subseteq \mathbb{R}^k\}$.

### 2.2 Deterministic Policy Gradients and TD-learning

Policy gradient methods improve the policy $\pi_{\boldsymbol{\theta}}$ by ascending the direction of the gradient of its performance function $J(\boldsymbol{\theta})$. The *Deterministic Policy Gradient Theorem* [47] provides a practical way to calculate this gradient. It shows that, under some mild regularity conditions on the MDP, the gradient of the performance of a deterministic policy $\pi_{\boldsymbol{\theta}}$ is given by:

$$\nabla_{\boldsymbol{\theta}} J(\boldsymbol{\theta}) = \frac{1}{1 - \gamma} \int_{\mathcal{S}} d_{\mu}^{\pi}(s) \nabla_a Q^{\pi}(s, a)\big|_{a = \pi_{\boldsymbol{\theta}}(s)} \nabla_{\boldsymbol{\theta}} \pi_{\boldsymbol{\theta}}(s) \mathrm{d}s. \tag{1}$$

This result can be interpreted through the lens of the chain rule applied to the *action-value gradient* $\nabla_{\boldsymbol{\theta}} Q^{\pi}$: the policy gradient does not directly depend on the gradient of $d_{\mu}^{\pi}$, and can be obtained by just chaining the actor's Jacobian $\nabla_{\boldsymbol{\theta}} \pi_{\boldsymbol{\theta}}$ with the *action-gradient* of the value function $\nabla_a Q^{\pi}$.

The theorem motivates a family of policy gradient actor-critic algorithms, such as DDPG [28] and TD3 [18]. Similarly to the classical policy iteration [52], the evaluation of a policy $\pi \in \Pi$ (called *actor* in this context) is interleaved with its improvement w.r.t the approximate action-value function $\widehat{Q} \in \mathcal{Q}$ (called *critic*). Specifically, the typical desideratum consists in finding a critic $\widehat{Q}$ which minimizes the *policy evaluation error*:

$$\widehat{Q} \in \underset{\widetilde{Q} \in \mathcal{Q}}{\arg\min} \ \underset{s \sim d_{\mu}^{\pi}}{\mathbb{E}} \left| \delta^{\pi, \widetilde{Q}}(s, \pi(s)) \right|, \tag{2}$$

where $\delta^{\pi, \widetilde{Q}}(s, a) = Q^{\pi}(s, a) - \widetilde{Q}(s, a)$ is a deviation w.r.t the true state-action value. Given the lack of knowledge about the transition model, $Q^{\pi}$ needs to be approximated. A common approximation technique consists in employing the *temporal difference* (TD) *error* [50], defined as $\widehat{\delta}^{\pi, \widetilde{Q}}(s, a, s') = r(s, a) + \gamma \widetilde{Q}(s', \pi(s')) - \widetilde{Q}(s, a)$, giving rise to a bootstrapped optimization criterion for $\widehat{Q}$:

$$\widehat{Q} \in \underset{\widetilde{Q} \in \mathcal{Q}}{\arg\min} \ \underset{\substack{s \sim d_{\mu}^{\pi} \\ s' \sim p(\cdot | s, \pi(s))}}{\mathbb{E}} \left| \widehat{\delta}^{\pi, \widetilde{Q}}(s, \pi(s), s') \right|. \tag{3}$$

Minimizing the TD-error, albeit under rather strong assumptions, enjoys convergence guarantees [52, 56]. Once a critic is learned, the actor $\pi_{\theta}$ can be improved by maximizing the action-value function for actions produced by the policy:

$$\pi_{\theta} \in \underset{\widetilde{\pi}_{\theta} \in \Pi_{\Theta}}{\arg\max} \underset{s \sim d_{\mu}^{\pi}}{\mathbb{E}} \left[ \widehat{Q}(s, \widetilde{\pi}_{\theta}(s)) \right]. \tag{4}$$

The above can be seen as a generalization of the policy improvement step in classical policy iteration, which relies on maximization over a discrete action space that cannot be easily carried out in continuous spaces. In practice, to reduce the computational burden, the problems in Equation 3 and Equation 4 are solved only partially (e.g., by using a single optimization step) at each iteration, similarly to generalized policy iteration [52].

## 3 Learning Action-Value Gradients

In this section, we discuss theoretically how to learn a useful critic in the context of deterministic policy gradients. Then, we make the theoretical insights concrete and, guided by practical considerations, present *Model-based Action-Gradient-Estimator Policy Optimization* (MAGE), a novel policy optimization algorithm.

### 3.1 How to Learn a Useful Critic?

An actor can only be as good as allowed by its critic. Thus, obtaining an *effective* critic is one of the most crucial passages for any actor-critic algorithm. In the previous section, we outlined the most common method to train the critic, consisting in the minimization of the temporal difference error. However, when the learned action-value function will not be perfect, as common in policy optimization with function approximation, minimizing the TD-error does not guarantee that the critic will be effective for the goal of solving the control problem. Instead, the following result provides foundations for a more grounded objective function for critic learning.

**Proposition 3.1.** *Let $\Pi$ be a parametric space of $L_{\pi}$-Lipschitz continuous differentiable deterministic policies, $\mathcal{Q}$ a space of approximate value functions and $\| \cdot \|$ any p-norm. Given $\pi \in \Pi$ and $\widehat{Q} \in \mathcal{Q}$, the norm of the difference between the true policy gradient $\nabla_{\theta} J(\theta)$ and its approximation $\widehat{\nabla}_{\theta} J(\theta)$, which uses $\widehat{Q}$, can be upper bounded as:*

$$\| \nabla_{\theta} J(\theta) - \widehat{\nabla}_{\theta} J(\theta) \| \leq \frac{L_{\pi}}{1 - \gamma} \underset{s \sim d_{\mu}^{\pi}}{\mathbb{E}} \left\| \nabla_a \delta^{\pi, \widehat{Q}}(s, a) \Big|_{a = \pi(s)} \right\|.$$

The proposition (see Appendix A for the proof) is a direct consequence of the *Deterministic Policy Gradient Theorem* and is thus valid when deterministic policies are employed. The Lipschitz assumption for $\pi$ is easily satisfied for many policy classes of practical use, e.g., neural networks [16].

Proposition 3.1 suggests that it is the norm of the action-gradient of the policy evaluation error instead of its value that should be minimized to reduce the bias introduced by the use of the approximate value function $\widehat{Q}$. To minimize the bound, a proxy for the unknown $Q^{\pi}$ is needed. To this aim, it is possible to follow the approach of traditional TD-learning, substituting the evaluation error $\delta^{\pi, \widehat{Q}}$ with the TD-error $\widehat{\delta}^{\pi, \widehat{Q}}$. This leads to the following optimization problem:

$$\widehat{Q} \in \underset{\widetilde{Q} \in \mathcal{Q}}{\arg\min} \underset{\substack{s \sim d_{\mu}^{\pi} \\ s' \sim p(\cdot | s, \pi(s))}}{\mathbb{E}} \left\| \nabla_a \widehat{\delta}^{\pi, \widetilde{Q}}(s, \pi(s), s') \right\|. \tag{5}$$

Notice that computing the gradient w.r.t. the action of the TD-error $\widehat{\delta}^{\pi, \widehat{Q}}$ requires taking into account the effect of action $a$ on the transition to the subsequent state in the environment $s'$, i.e., backpropagating through the environment dynamics $p$. Since $p$ is not available in typical RL settings, especially in a differentiable form, it needs to be substituted with an approximate model $\widehat{p}$, as commonly done in model-based RL [7, 10, 23]. An environment model gives rise to imaginary transitions $(s, \pi(s), \widehat{s})$, where $\widehat{s} \sim \widehat{p}(\cdot | s, \pi(s))$. Given differentiable model, policy, and action-value function, the action-gradient can be effectively computed by leveraging standard automatic differentiation

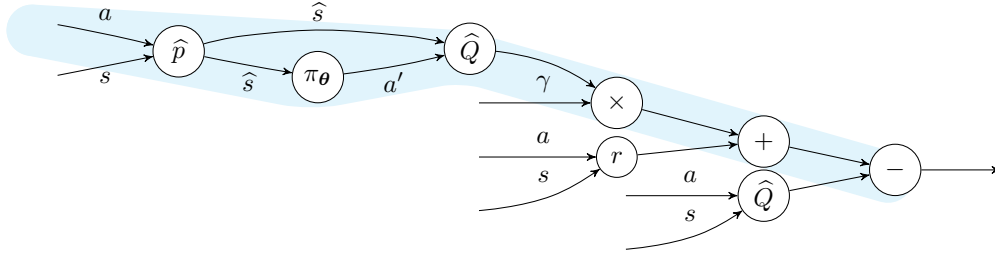

Figure 1: Graph describing the computation of $\widehat{\delta}^{\pi_{\boldsymbol{\theta}},\widehat{Q}}$, when using policy $\pi_{\boldsymbol{\theta}}$, model $\widehat{p}$, action-value function $\widehat{Q}$. Nodes and edges represent functions and variables, respectively. To compute $\nabla_a\widehat{\delta}^{\pi_{\boldsymbol{\theta}},\widehat{Q}}$, all the paths from the output back to $a$ must be considered, including the one highlighted in cyan, which involves the environment dynamics. Therefore, an approximate differentiable model $\widehat{p}$ needs to be learned in order to make all the required paths accessible.

tools [5]. The corresponding computational graph is depicted in Figure 1. This leads to a viable way to obtain $\widehat{Q}$:

$$\widehat{Q} \in \underset{\widetilde{Q}\in\mathcal{Q}}{\arg\min} \ \underset{\substack{s \sim d_\mu^\pi \\ \widehat{s} \sim \widehat{p}(\cdot|s,\pi(s))}}{\mathbb{E}} \left\| \nabla_a \widehat{\delta}^{\pi,\widetilde{Q}}(s,\pi(s),\widehat{s}) \right\|. \tag{6}$$

Even in the general case of a stochastic model, differentiating through the resulting computations is still possible for many commonly used model classes via the reparametrization trick [21]. Using an approximate model $\widehat{p}$ implies a tradeoff, since additional bias is injected into the estimation of the critic. Nonetheless, the use of $\widehat{p}$ is the most direct way to solve the optimization problem in Equation 6 and to obtain a $\widehat{Q}$ that provides a more accurate policy gradient w.r.t. the typical critic.

## 3.2 Model-based Action-Gradient-Estimator Policy Optimization

The outlined procedure for learning the value function requires an approximate model $p_{\boldsymbol{\omega}}$, thus naturally suggesting its integration into a model-based policy optimization framework. A model-based actor-critic method involves three steps during each iteration: learning the model $p_{\boldsymbol{\omega}}$, updating the action-value function $Q_{\boldsymbol{\phi}}$ and improving the policy $\pi_{\boldsymbol{\theta}}$. In the following, we consider neural networks as function approximators to represent the three modules, although any class of differentiable models could be leveraged. Our approach is inspired by Dyna [51], and employs an approximate dynamics model for generating 1-step imaginary on-policy transitions starting from observed states stored in a replay buffer. Those transitions are then employed to learn $Q_{\boldsymbol{\phi}}$, and, in turn, leveraged for computing an improvement direction for the parameters of the policy $\pi_{\boldsymbol{\theta}}$.

In preliminary experiments, we found that directly solving the minimization problem in Equation 6 is hard in practice. During the optimization, the parameters are prone to be trapped in local-minima, which leads to degenerate solutions. A demonstration of this phenomenon is detailed in Appendix B.1. The root cause of this effect is unknown and suggests the existence of a tradeoff between the easier minimization of the TD-error and the more theoretically grounded minimization of its action-gradient.

We propose as a remedy the introduction of a constraint into the optimization problem. We argue that, among the possible solutions, a natural one is constraining the optimization landscape by bounding the traditional TD-error (see Equation 3), and thus solving the following optimization problem:

$$\begin{aligned}
&\underset{\widetilde{\phi}\in\Phi}{\min} \ \underset{\substack{s \sim d_\mu^\pi \\ \widehat{s} \sim p_{\boldsymbol{\omega}}(\cdot|s,\pi(s))}}{\mathbb{E}} \left\| \nabla_a \widehat{\delta}^{\pi,Q_{\widetilde{\phi}}}(s,a,\widehat{s}) \Big|_{a=\pi(s)} \right\| \\
&\text{s.t.} \ \underset{\substack{s \sim d_\mu^\pi \\ \widehat{s} \sim p_{\boldsymbol{\omega}}(\cdot|s,\pi(s))}}{\mathbb{E}} \left| \widehat{\delta}^{\pi,Q_{\widetilde{\phi}}}(s,\pi(s),\widehat{s}) \right| \leq \lambda.
\end{aligned} \tag{7}$$

As the above expressions already require non-trivial gradient computations, we avoid the use of complex and expensive methods for nonlinear programming. Instead, we resort to *penalty function*

**Algorithm 1** Model-based Action-Gradient-Estimator Policy Optimization (MAGE)

---
**Input:** Initial buffer $\mathcal{B}$, set of parameter vectors $\{\boldsymbol{\omega}, \boldsymbol{\phi}, \boldsymbol{\theta}\}$
**for** each iteration **do**
    Collect transition $(s, a, s')$ acting according to exploratory version of $\pi_{\boldsymbol{\theta}}$
    $\mathcal{B} \leftarrow \mathcal{B} \cup \{(s, a, s')\}$
    **for** each model learning step **do**
        $\boldsymbol{\omega} \leftarrow \boldsymbol{\omega} - \alpha_p \nabla_{\boldsymbol{\omega}} \ell(s, a, s'; \boldsymbol{\omega}), \qquad (s, a, s') \sim \mathcal{B}$
    **end for**
    **for** each policy optimization step **do**
        Extract state $s$ after sampling $(s, \cdot, \cdot) \sim \mathcal{B}$
        $\bar{\boldsymbol{\phi}} \leftarrow \boldsymbol{\phi}$
        $\widehat{\delta}(s, a, \widehat{s}; \boldsymbol{\phi}) \leftarrow r(s, a) + \gamma Q_{\bar{\boldsymbol{\phi}}}(\widehat{s}, \pi_{\boldsymbol{\theta}}(\widehat{s})) - Q_{\boldsymbol{\phi}}(s, a), \qquad a = \pi_{\boldsymbol{\theta}}(s), \widehat{s} \sim p_{\boldsymbol{\omega}}(\cdot | s, a)$
        $\boldsymbol{\phi} \leftarrow \boldsymbol{\phi} - \alpha_Q \nabla_{\boldsymbol{\phi}} \left( \left\| \nabla_a \widehat{\delta}(s, a, \widehat{s}; \boldsymbol{\phi}) \big|_{a=\pi_{\boldsymbol{\theta}}(s)} \right\| + \lambda \left| \widehat{\delta}(s, a, \widehat{s}; \boldsymbol{\phi}) \right| \right)$
        $\boldsymbol{\theta} \leftarrow \boldsymbol{\theta} + \alpha_\pi \nabla_{\boldsymbol{\theta}} Q_{\boldsymbol{\phi}}(s, \pi_{\boldsymbol{\theta}}(s))$
    **end for**
**end for**

---

*methods* [49] by regularizing the original objective by using the TD-error. A similar approach has been used in the past in, e.g., Proximal Policy Optimization (PPO, [45]) to approximately solve different constrained optimization problems.

Eventually, the parameters of $Q_{\boldsymbol{\phi}}$ are learned by descending the gradient

$$\nabla_{\boldsymbol{\phi}} \mathcal{L}(s, a, \widehat{s}; \boldsymbol{\phi}, \boldsymbol{\theta}, \boldsymbol{\omega}) = \nabla_{\boldsymbol{\phi}} \left( \left\| \nabla_a \widehat{\delta}^{\pi_{\boldsymbol{\theta}}, Q_{\boldsymbol{\phi}}}(s, a, \widehat{s}) \big|_{a=\pi_{\boldsymbol{\theta}}(s)} \right\| + \lambda \left| \widehat{\delta}^{\pi_{\boldsymbol{\theta}}, Q_{\boldsymbol{\phi}}}(s, a, \widehat{s}) \right| \right) \qquad (8)$$

on an imaginary transition $(s, a, \widehat{s})$. This expression requires computing second-order gradients, which would be computationally expensive if computed w.r.t. to the high-dimensional space of parameters $\Phi$ of the $Q$-function. Here, however, the optimization is affordable since the gradients are computed w.r.t., typically low dimensional, actions. Notice also that the computational overhead of the second term in Equation 8 is minimal, since evaluating the TD-error $\widehat{\delta}^{\pi, Q_{\boldsymbol{\phi}}}(s, a, \widehat{s})$ is anyway, when using automatic differentiation, required to compute its gradient.

We plug our critic training method into a model-based Dyna-like algorithm, giving rise to *Model-based Action-Gradient-Estimator Policy Optimization* (MAGE), which is presented[2] in Algorithm 1. At each iteration, the dynamics model $p_{\boldsymbol{\omega}}$ is trained to maximize the likelihood of the transitions stored in the experience replay buffer $\mathcal{B}$, or, equivalently, to minimize an appropriate loss function $\ell$:

$$\boldsymbol{\omega} \in \underset{\widetilde{\boldsymbol{\omega}} \in \Omega}{\arg\min} \; \underset{(s, a, s') \sim \mathcal{B}}{\mathbb{E}} \left[ \ell(s, a, s'; \widetilde{\boldsymbol{\omega}}) \right]. \qquad (9)$$

Then, for one or more steps, the TD-error for the current policy and action-value function is computed, and used together with its action-gradient to update $Q_{\boldsymbol{\phi}}$, which in turn is leveraged to improve $\pi_{\boldsymbol{\theta}}$.

## 4 Experiments

### 4.1 Sample-Efficient Continuous Control with MAGE

**Algorithm settings** The general structure of MAGE is compatible with many actor-critic algorithms with deterministic policies. In this experiment, we employ TD3 [18], a popular, state-of-the-art extension to DDPG [28], as a base policy optimization method. This amounts to the addition of target policy smoothing, delayed policy updates, clipped double Q-learning and target functions. We call this version of our algorithm MAGE-TD3[3]. After each step of environment interaction, we add the collected transition in the replay buffer $\mathcal{B}$, train the approximate model $p_{\boldsymbol{\omega}}$, and update critic and actor 10 times. We employ a single value of $\lambda = 0.2$ for all the environments, since we found MAGE to be reasonably robust to the choice of this hyperparameter (see Appendix B). In order to reduce the impact of model bias, MAGE leverages an ensemble of 8 probabilistic Gaussian-output models, trained by maximum likelihood estimation.

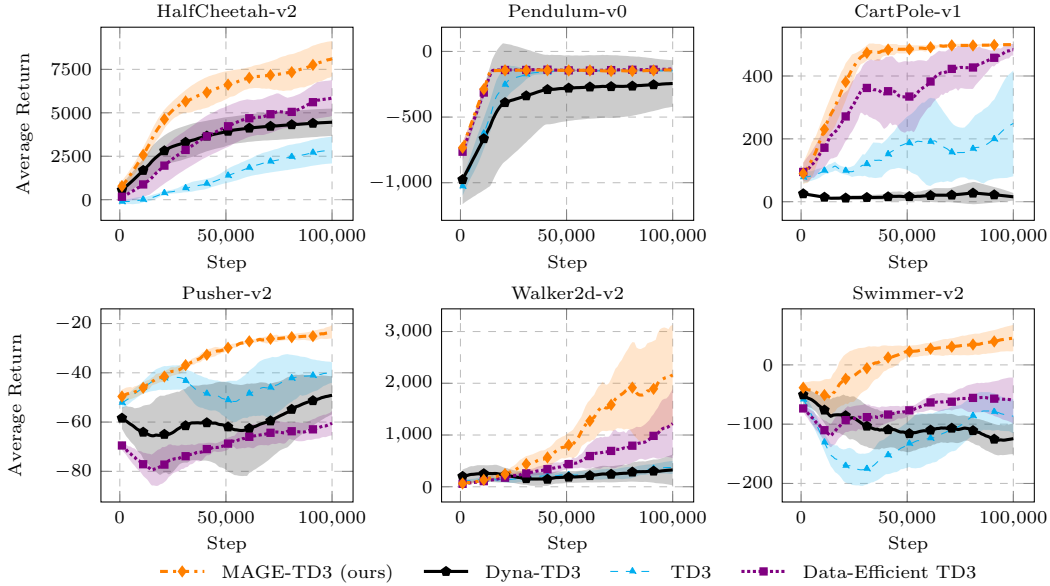

Figure 2: Performance in terms of average return of MAGE on continuous control benchmarks. MAGE compares favorably to the three baselines on all the environments (5 runs, 95% c.i.).

**Baselines and environments**   We consider one model-based and two model-free algorithms as baselines. The first one is Dyna-TD3, which uses a classical TD-error loss, otherwise being identical to MAGE-TD3. It resembles 1-step horizon *Model-based Policy Optimization* (MBPO [23]), but uses a deterministic policy optimized by TD3. Apart from that, we compared MAGE against TD3 and its sample-efficient variant [57], which employs multiple updates for each environment step and trades off computational efficiency and, potentially, stability [32] for sample efficiency. Specifically, for a fair comparison with MAGE-TD3, we execute 10 critic and actor updates after each interaction with the environment. We employ environments from OpenAI Gym [6] and the MuJoCo physics simulator [55] as continuous control benchmarks, assuming, for all the environments, the availability of a differentiable reward function (we will later show that MAGE behaves well also in the absence of this assumption). Additional details concerning the experimental setting are reported in Appendix D.

**Results**   Figure 2 shows the learning curves for the average return of all the approaches. Since our primary interest is MAGE's sample efficiency, we show the first $10^5$ steps of environment interaction. The results show that MAGE is able to learn at least as fast as all the baselines on all the environments, confirming the intuitive advantage of directly optimizing for the accuracy of the estimated action-value gradient. Interestingly, no superiority of the vanilla Dyna-TD3 on its simple data-efficient version can be observed: this demonstrates that there is no intrinsic advantage in terms of sample-efficiency for model-based reinforcement learning, but it is instead highly environment- and algorithm-dependent. On the other hand, increasing the number of offline updates for model-free algorithms can hurt performance in some environments, as it is the case, for instance, on the Pusher-v2 environment. Note that, in contrast with Dyna-TD3, that only leverages the model as a generator for additional transitions w.r.t. the ones that can be obtained in the environment, MAGE makes deeper use of the learned model of the dynamics in order to unlock a peculiar learning modality that would be impossible in a model-free setting. In Appendix B, we also show that MAGE-TD3 matches the asymptotic performance of its model-free counterpart.

## 4.2   Understanding MAGE

**Action-Gradient Estimation**   MAGE was designed to obtain a critic that is maximally useful for policy improvement by yielding accurate action-value gradients. How much better does it predict them compared to the traditional TD-learning? To investigate this question, we employ the Pendulum-v0 environment, using a differentiable oracle in place of the approximate dynamics model.

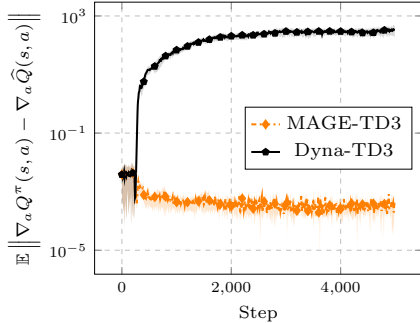

Figure 3: Error of critics in predicting $\nabla_a Q^\pi$ for a random $\pi$ (4, runs, 95% c.i.). Notice the log scale on the Y axis.

We fix a randomly initialized actor, and train only its critic with both MAGE-TD3 and its Dyna counterpart. During training, for each transition on a trajectory, we compute the true action-gradient as $\nabla_a Q^\pi(s_t, a_t) = \nabla_a \sum_{t'=t}^{H-1} \gamma^{t'} r(s_{t'}, a_{t'})|_{a_{t'}=\pi_\theta(s_{t'})}$ and compare it to the action-gradient $\nabla_a \widehat{Q}$ provided by the learned critic. The results, shown in Figure 7, indicate that the MAGE's critic progressively learns an accurate estimate of the action-gradient; by contrast, the one trained using traditional temporal difference completely fails in predicting it. The results undermine the common assumption that minimizing the TD-error yields also a minimization of the error on the gradients. The difference can explain the superior sample efficiency of MAGE over classical TD-learning. We believe the surprising observation that traditional approaches are able to learn a reasonably good policy even when the learned gradient is very different from the real one is in line with recent analyses on the mismatch between the empirical behavior of policy gradient approaches and their conceptual features [22].

**Reward Availability** Throughout the presentation and evaluation of MAGE, we assumed complete knowledge of the reward function $r$ of the underlying Markov Decision Process. While this assumption is natural in many real-world settings [10] and thus commonly employed in other model-based reinforcement learning methods [7, 11, 21], its role is particularly crucial in our algorithm. In traditional temporal difference learning, given a transition $(s, a, \widehat{s})$, the reward $r(s, a)$ constitutes the only grounding element in the objective function. The reward function plays an even stronger role as a grounding element for bootstrapping in MAGE, since both its value $r(s, a)$ and its action-gradient $\nabla_a r(s, a)$ are needed: while the former can be usually observed in the environment, the latter can only be computed with complete knowledge of the underlying function. In our experiments on the sample efficiency of MAGE, we employed the ground-truth reward function (with ground-truth gradients): a natural question is whether MAGE still performs reasonably well if an estimated reward function $\hat{r}$, learned from data, is used in place of the real $r$. To

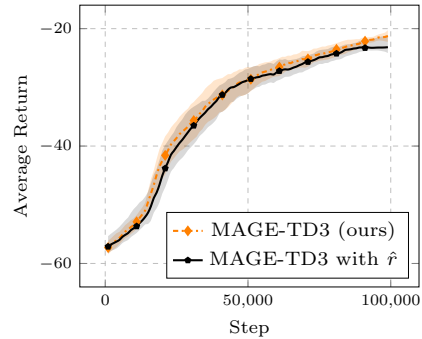

Figure 4: Performance of MAGE on Pusher-v2 using an estimated reward function $\hat{r}$ (5 runs, 95% c.i.).

answer this question, we evaluate a version of MAGE in which an approximate reward function $\hat{r}$ is learned by using a neural network approximator and minimizing the mean squared error on the rewards observed in the environment. The results, perhaps surprising, are reported in Figure 4 for the Pusher-v2 environment (see Appendix B.2 for the complete results). They show that, for the commonly employed continuous control benchmarks, the performance of our method is only minimally degraded by the use of an approximate reward function in place of the real one, thus suggesting inherent robustness to inaccurate evaluations of the reward function as well as its action-gradient.

## 5 Related Work

Policy gradients are among the most popular methods in reinforcement learning. A variety of algorithms have been proposed for the estimation of the policy gradient, either involving only the policy [4, 32, 61] or also a value function [33, 44, 45]. The latter category of algorithms is reffered to as *actor-critic* methods [26, 35]. Among them, the ones based on the *Deterministic Policy Gradient* [28, 47] leverage the action-gradient of the critic. When using function approximation, the quality of the learned critic is of paramount importance [3]: for instance, enforcing on the critic the *compatiblity conditions* [47] ensures an unbiased estimate of the policy gradient.

Developed around such conditions, GProp [2] is, to the best of our knowledge, the only method that explicitly optimizes for the accuracy of the learned action-value gradient. It is significantly

different w.r.t. MAGE, being model-free and based on gradient estimation via noisy perturbations together with an additional deviator network. Importantly, while GProp's deviator network is a function approximator that outputs an estimate for the action-gradient, recent theoretical [39] and practical [40] insights outside of RL suggest that learning the action-gradient by second-order differentiation, as we propose in MAGE, is not only simpler to implement w.r.t. GProp and similar procedures [59], but also fundamentally more effective when using neural network approximators.

The technique we use for learning the action-gradient relies on the differentiation of the TD-error and, thus, of the Bellman equation. This is related to a broad class of methods called *value gradients* [8, 14, 21, 41], in which the policy is improved by backpropagating through the unrolled Bellman equation. Those approaches, however, learn the value function by standard temporal difference [21]. Another classical method, named *Dual Heuristic Programming* (DHP) [13, 35, 60], learns the gradient of the state-value function in a model-based setting, leading to a TD-learning procedure that resembles our approach. However, DHP has the main goal of improving generalization of the value function and exploration, and is fundamentally different from MAGE, that aims at learning an accurate action-gradient of the critic and is motivated by the Deterministic Policy Gradient Theorem.

More broadly, inside and outside of reinforcement learning, several algorithms incorporate gradient penalties into the loss function used for training a neural network. This technique, known as *double backpropagation* [12], has been employed in a number of applications, for instance increasing generalization capabilities [12, 38], enforcing Lipschitz constants [19, 20, 30], or encouraging robustness to adversarial examples [48]. Particularly related to our approach is *Sobolev training* [9], which leverages the availability of the derivatives of a target function to explicitly try to learn both value and gradient of it during supervised training; in our case, no ground-truth gradient is available and we use the action-gradient of the TD-target as a proxy.

Our method learns the action-gradient in the context of model-based policy optimization [10, 54, 58]. We build upon the classical Dyna framework [25, 51], in which a learned model is used for generating imaginary transitions, then employed for training a value function. Our algorithm, which learns a Q-function from model-generated data but only optimizes the policy by using real data, is related to the approaches that compute the policy gradient by using a model-based value function together with trajectories sampled in the environment [1, 11, 21, 23]. In practice, we leverage an ensemble of models, which has been shown to improve performance in a variety of contexts [7, 23, 27].

Finally, our work is related in spirit to *decision-aware model learning* (DAML) [11, 15, 24]. In DAML approaches, the model of the dynamics of the environment is learned by explicitly considering how it will be used for improving the control policy: this is the same rationale behind the learning objective used in MAGE for the critic, focused on how it will be useful for policy optimization, and not merely on how it will be similar to the true value function.

# 6 Conclusion

In this paper, we presented MAGE, a model-based actor-critic algorithm with deterministic actor, which leverages an approximate dynamics model to directly learn the action-value gradient via temporal difference learning. MAGE employs second-order differentiation to obtain a critic tailored for policy improvement. The empirical evaluation of MAGE demonstrated its superiority over model-based and model-free baselines on challenging high-dimensional continuous control tasks.

A limitation of our method is of computational nature: in addition to the cost of model learning paid also by other model-based actor-critic algorithms, we incur the expense of computing a second-order gradient in order to train the critic, in result, approximately doubling the training time in comparison to the Dyna-based policy gradient approach. This can potentially be alleviated by the development of more efficient automatic differentiation tools, which is, currently, an active area of research [5].

While it is often hard to determine the circumstances under which the addition of an approximate model to a model-free algorithm is beneficial [23], we have shown that model-based techniques, such as MAGE's gradient-learning procedure, can unlock novel learning modalities, otherwise inaccessible. This can actually be the true power of model-based reinforcement learning. Therefore, apart from improving MAGE (e.g., by investigating the unconstrained critic learning problem) and generalizing it (e.g., to value gradients with real trajectories [21] or multi-steps methods [17]), we hope that future work will reveal other innovative learning schemes that are infeasible in model-free settings.

## Broader Impact

The method presented in this paper is a reinforcement learning algorithm that can be used to control a system executing real-valued actions in an environment. Therefore, a natural application of it is in robotics, with positive (e.g., elderly care, resource-efficiency in manufacturing) and negative (e.g., military) applications. Alongside other deep reinforcement learning algorithms, our method is computationally intensive and its training can thus require considerable resources (i.e., hardware and electricity); on the other hand, given that in many real-world scenarios every interaction with a system implies an economic or environmental cost, the sample efficiency of MAGE is aligned with the modern principles of responsible artificial intelligence.

## Acknowledgments and Disclosure of Funding

The authors are grateful to Miroslav Štrupl for discussions about the relationship between action-gradients and TD-learning, David Alvarez for co-authoring the reinforcement learning framework used for the experiments, Christian Osendorfer, Miroslav Štrupl, Jan Koutník for their valuable feedback on an early draft of this manuscript, and to everyone at NNAISENSE for contributing to an inspiring research environment. This work was funded by NNAISENSE S.A..

## Footnotes

[2]For simplicity of presentation, an abstract version of MAGE is considered in Algorithm 1. Any actor-critic algorithm with deterministic actor can be then used to instantiate MAGE into a practical incarnation.

[3]The PyTorch [34] implementation, based on [46], is available at `https://github.com/nnaisense/MAGE`.

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
