[Supplementary Material]

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

_{\boldsymbol{\theta}} \in \arg\max_{\widetilde{\pi}_{\boldsymbol{\theta}} \in \Pi_{\Theta}} \mathbb{E}_{s \sim d_{\mu}^{\pi}} \left[ \widehat{Q}(s, \widetilde{\pi}_{\boldsymbol{\theta}}(s)) \right]. \tag{4}$$

The above can be seen as a generalization of the policy improvement step in classical policy iteration, which relies on maximization over a discrete action space that cannot be easily carried out in continuous spaces. In practice, to reduce the computational burden, the problems in Equation 3 and Equation 4 are solved only partially (e.g., by using a single optimization step) at each iteration, similarly to generalized policy iteration [52].

## 3 Learning Action-Value Gradients

In this section, we discuss theoretically how to learn a useful critic in the context of deterministic policy gradients. Then, we make the theoretical insights concrete and, guided by practical considerations, present *Model-based Action-Gradient-Estimator Policy Optimization* (MAGE), a novel policy optimization algorithm.

### 3.1 How to Learn a Useful Critic?

An actor can only be as good as allowed by its critic. Thus, obtaining an *effective* critic is one of the most crucial passages for any actor-critic algorithm. In the previous section, we outlined the most common method to train the critic, consisting in the minimization of the temporal difference error. However, when the learned action-value function will not be perfect, as common in policy optimization with function approximation, minimizing the TD-error does not guarantee that the critic will be effective for the goal of solving the control problem. Instead, the following result provides foundations for a more grounded objective function for critic learning.

**Proposition 3.1.** *Let $\Pi$ be a parametric space of $L_{\pi}$-Lipschitz continuous differentiable deterministic policies, $\mathcal{Q}$ a space of approximate value functions and $\| \cdot \|$ any p-norm. Given $\pi \in \Pi$ and $\widehat{Q} \in \mathcal{Q}$, the norm of the difference between the true policy gradient $\nabla_{\boldsymbol{\theta}} J(\boldsymbol{\theta})$ and its approximation $\widehat{\nabla}_{\boldsymbol{\theta}} J(\boldsymbol{\theta})$, which uses $\widehat{Q}$, can be upper bounded as:*

$$\| \nabla_{\boldsymbol{\theta}} J(\boldsymbol{\theta}) - \widehat{\nabla}_{\boldsymbol{\theta}} J(\boldsymbol{\theta}) \| \leq \frac{L_{\pi}}{1 - \gamma} \mathbb{E}_{s \sim d_{\mu}^{\pi}} \

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

# A   Proof of Proposition 3.1

The proof follows directly from the Deterministic Policy Gradient Theorem, Therefore, the Proposition inherits all of its smoothness assumptions about the Markov Decision Process [47].

**Proposition 3.1.** *Let $\Pi$ be a parametric space of $L_\pi$-Lipschitz continuous differentiable deterministic policies, $\mathcal{Q}$ a space of approximate value functions and $\|\cdot\|$ any $p$-norm. Given $\pi \in \Pi$ and $\widehat{Q} \in \mathcal{Q}$, the norm of the difference between the true policy gradient $\nabla_{\boldsymbol{\theta}} J(\boldsymbol{\theta})$ and its approximation $\widehat{\nabla}_{\boldsymbol{\theta}} J(\boldsymbol{\theta})$, which uses $\widehat{Q}$, can be upper bounded as:*

$$\|\nabla_{\boldsymbol{\theta}} J(\boldsymbol{\theta}) - \widehat{\nabla}_{\boldsymbol{\theta}} J(\boldsymbol{\theta})\| \leq \frac{L_\pi}{1-\gamma} \underset{s \sim d_\mu^\pi}{\mathbb{E}} \left\| \nabla_a \delta^{\pi,\widehat{Q}}(s,a) \Big|_{a=\pi(s)} \right\|.$$

*Proof.*

$$\|\nabla_{\boldsymbol{\theta}} J(\boldsymbol{\theta}) - \widehat{\nabla}_{\boldsymbol{\theta}} J(\boldsymbol{\theta})\| = \frac{1}{1-\gamma} \left\| \int_{\mathcal{S}} d_\mu^\pi(s) \left( \nabla_a Q^\pi(s,a)|_{a=\pi(s)} - \nabla_a \widehat{Q}(s,a)|_{a=\pi(s)} \right) \nabla_{\boldsymbol{\theta}} \pi(s) \mathrm{d}s \right\| \tag{10}$$

$$= \frac{1}{1-\gamma} \left\| \int_{\mathcal{S}} d_\mu^\pi(s) \nabla_a \delta^{\pi,\widehat{Q}}(s,a)|_{a=\pi(s)} \nabla_{\boldsymbol{\theta}} \pi(s) \mathrm{d}s \right\| \tag{11}$$

$$\leq \frac{1}{1-\gamma} \int_{\mathcal{S}} d_\mu^\pi(s) \left\| \nabla_a \delta^{\pi,\widehat{Q}}(s,a)|_{a=\pi(s)} \right\| \cdot \|\nabla_{\boldsymbol{\theta}} \pi(s)\| \, \mathrm{d}s$$

$$\leq \frac{L_\pi}{1-\gamma} \int_{\mathcal{S}} d_\mu^\pi(s) \left\| \nabla_a \delta^{\pi,\widehat{Q}}(s,a)|_{a=\pi(s)} \right\| \mathrm{d}s. \tag{12}$$

$\square$

Equation 10 follows from the Deterministic Policy Gradient Theorem. To obtain Equation 11, we exploit the definition of $\delta^{\pi,\widehat{Q}}$ and linearity of differentiation. Finally, in Equation 12, we use the Lipschitz policy assumption.

# B   Additional Experiments

## B.1   Unconstrained Action-Value Gradient learning

Proposition 3.1 directly encourages training the critic by minimizing the bound on the error of the policy gradient, i.e., the norm of the action-gradient of the policy evaluation error.

Figure 5: Action-gradients

However, we found a direct optimization of this bound, by means of the TD-error, difficult in the context of Dyna-like algorithms. We analyze this behavior in the Pendulum-v0 environment [6], instantiating a version of MAGE based on DDPG [28] (MAGE-DDPG). To understand the learning dynamics of the action-value gradients in a way that is not affected by the model bias, we employ the differentiable version of the real environment dynamics and test MAGE without the TD-error regularization (i.e., with $\lambda = 0$). Therefore, at each step, $Q_{\boldsymbol{\omega}}$ is improved by minimizing the norm of $\widehat{\delta}$ computed on transitions whose next state is sampled from $p$. Unfortunately, no useful learning can be achieved in this setting: a degenerate solution consisting of $\widehat{Q}$ such that $\left\| \nabla_a \widehat{Q}(s,a) \right\| \approx 0, \forall s \in \mathcal{S}, \forall a \in \mathcal{A}$ is rapidly reached, as shown in Figure 5. We employ exactly the settings and hyperparamters that are successfully employed in the full version of MAGE.

We believe that understanding whether, or under which circumstances, the direct minimization of the bound in Proposition 3.1 is possible is an interesting open question.

Figure 6: Performance in terms of average return of MAGE-TD3 and Dyna-TD3 with and without the use of an estimated reward function $\hat{r}$ (5 runs, 95% c.i.).

## B.2 MAGE with Trained Reward Function

As discussed in Section 4, MAGE is able to achieve good performance even with an estimated reward function. We report in Figure 6 the full results of this experiment on all the considered environments. For reference, we test MAGE and Dyna-TD3 as well as their versions in which the ground-truth reward function is substituted with one trained on the experience replay data using the MSE loss.

The results indicate that learning the reward function when it is not directly accessible does not produce any catastrophic harm to the performance of the algorithm. Therefore, our approach remains competitive even when the assumption of a know differentiable reward function is not satisfied.

## B.3 Importance of model capacity

The quality of the learned model is of paramount importance for most MBRL algorithms, whose performance, generally, deteriorates when the model is not enough expressive for a given task. Thus, we performed an additional experiment to investigate how the performance of MAGE (compared to the Dyna-TD3 baseline) is affected by the use of less powerful models. We evaluated two versions of the model with reduced capacity: (i) only 2 members in the ensemble, 4 hidden layers and 256 units per layer (*-small suffix*); (ii) no ensemble (a single model), 2 hidden layers, 256 units per layer (*-smaller suffix*). Recall that the original setting involves a more powerful model with 8 members in the ensemble, 4 hidden layers, 512 units per layer (no suffix). The results on the Pusher-v2 environment, reported in Figure 7, show that MAGE is robust to the presence of a misspecified model: while a simpler but still quite capable model does no harm to MAGE, a significantly smaller model has a reasonable impact on the obtained average return.

Figure 7: Pusher-v2 (5 runs, 95% c.i.).

## B.4 Importance of $\lambda$

Our practical solution to viably minimize the norm of the action-gradient of the TD-error involves a constrained optimization problem, that limits the magnitude of the traditional TD-error. We approximately solve this problem by transforming it into an unconstrained one, introducing a new hyperparameter $\lambda$. $\lambda$ can be seen as a weight that is given to the traditional TD-error, assigning more or less importance to it compared to the error on the action-gradient. In the main experiment shown in the paper, we used $\lambda = 0.2$, which was chosen arbitrarily. How sensitive is MAGE to this parameter?

To study that, we carried out an experiment on the environment HalfCheetah-v2, by testing the TD3-based version of MAGE using four different values of $\lambda$. The results are shown in Figure 8, and demonstrate that, regardless of the value of $\lambda$, MAGE is significantly better than the baseline Dyna-TD3. MAGE is therefore robust to the choice of this hyperparameter. Notice also that the $\lambda = 0.2$ we used is probably not optimal for some environments: thus, the absolute returns obtained by MAGE could be improved

Figure 8: Median return of MAGE for different $\lambda \in [0, 1]$ (5 runs, 95% c.i.).

for particular tasks if a different hyperparameter is used for each of them, which we leave for a future work. Nonetheless, we decided to report in Figure 2 results for a fixed value of $\lambda$ across all the environments to show the robustness and ease of use of MAGE.

## B.5 Asymptotic Performance

A particularly important question concerning the performance of a model-based reinforcement learning algorithm is whether it matches the one of model-free baselines. We answer this question by running MAGE-TD3 and Dyna-TD3 until convergence on the HalfCheetah-v2 environment and replicating the evaluation procedure used to obtain the asymptotic performance of an algorithm in the original TD3 paper [18]. This amounts to repeating a number of trials (we use 5 of them for both the model-based algorithms) and considering the maximum average return over them.

Table 1: Maximum Average Return on HalfCheetah-v2 over different trials for three versions of TD3. Best performance is in bold.

| Algorithm | Maximum Average Return |
|---|---|
| MAGE-TD3 ($3 \cdot 10^6$ steps, 5 trials) | **9660.79 $\pm$ 2821.31** |
| Dyna-TD3 ($3 \cdot 10^6$ steps, 5 trials) | 8372.76 $\pm$ 1859.73 |
| TD3 ($1 \cdot 10^7$ steps, 10 trials) | 9636.95 $\pm$ 859.07 |

Results are reported in Table 1. Two conclusions can be drawn from them. The first one is that MAGE not only shows superior sample-efficiency compared to its model-free counterpart, but also matches its asymptotic performance in a smaller number of steps (less than one third). Secondly, the inferior performance of Dyna-TD3 compared to model-free TD3 once again reinforces the evidence that any simple introduction of a dynamics model into a model-free algorithm does not guarantee an improvement, when measuring performance using commonly employed metrics.

## C Action-Gradient of the TD-error

In this section, we present some additional information about the computation of the action-gradient of the TD-error, carried out during the critic learning step of MAGE. To implement MAGE, we employed PyTorch [34] and its automatic differentiation tools in order to compute the second-order gradient required by our method. In this way, we did not need to explicitly derive a closed form expression for a given model class or neural network architecture. Nonetheless, we report here the

Figure 9: Alternative view of the computational graph constructed during the computation of the TD-error $\widehat{\delta}$, following the notation from [43]. Round nodes represent stochastic variables, squares represent deterministic variables. Nodes with incoming dashed edges also depend on the state $s$.

general expression for the action-gradient of the TD-error:

$$\frac{\partial \widehat{\delta}^{\pi,\widehat{Q}}(s,a,s')}{\partial a} = \frac{\partial r(s,a)}{\partial a} + \gamma \frac{\partial \widehat{p}(s'|s,a)}{\partial a} \left( \frac{\partial \widehat{Q}(s',\pi(s'))}{\partial s'} + \frac{\partial \pi(s')}{\partial s'} \frac{\partial \widehat{Q}(s',a')}{\partial a'} \right) - \frac{\partial \widehat{Q}(s,a)}{\partial a}.$$

(13)

In MAGE, we employ a Gaussian stochastic model $\widehat{p}$: therefore, its action-gradient $\frac{\partial \widehat{p}(s'|s,a)}{\partial a}$ can be obtained by reparameterizing this distribution using randomly drawn unit Gaussian noise together with the learned mean and standard deviations. In our experiments, we only deal with continuous state and action spaces; however, by leveraging appropriate approximations (e.g., concrete distributions [31]), similar techniques can be employed also in the case of a discrete state space $\mathcal{S}$.

To further visualize the constructed computational graph, it is possible to employ a different view, inspired recent work on stochastic computational graphs [43], w.r.t. the one leveraged in Figure 1 (see Figure 9). In our case, the only possibly stochastic entity is the approximate model.

## D  Experimental details

### D.1  Instantiating MAGE

We presented in Algorithm 1 a generic version with MAGE, whose structure can be adapted to many model-free actor-critic algorithms. In most of our experiments, we use TD3 [18] as a reference algorithm, due to its stability and performance, giving birth to MAGE-TD3. In Algorithm 2, we report pseudocode for this version of our method. Unfortunately, while the use of the model is unchanged w.r.t. the abstract version, the addition of a second value function implies the computational overhead of using second-order differentiation twice.

### D.2  Hyperparameters

We employ 1000 (100 for the Pendulum and Cartpole environments) warmup steps of interaction with the environment before starting to update the critic and the actor. We use an ensemble of $8$ neural network as approximate dynamics models, that learn both mean and standard deviation of a Gaussian distribution, of 4 hidden layers of $512$ neurons (2 layers with $128$ units for the Pendulum and Cartpole environments) with swish [37] activation function. They are trained by maximum likelihood, minimizing the loss function, after every 25 steps of interaction with the environment, on 120 batches of 256 samples. We employ multi-layer perceptrons also for the actor (2 layers, $128$ neurons each for the Pendulum and CartPole environments and $284$ for all the others) and the critic (2 layers, $384$ neurons each). Model, actor and critic are trained with the RAdam optimizer [29], with learning rates of $0.0001$ and default parameters, and a weight decay of $0.0001$ for the approximate dynamics model. For MAGE-TD3, we employ $\lambda = 0.2$ for the experiment showed in Figure 2 and $\lambda = 0.05$ for the other experiments. We update the critic and the actor by extracting 1024 (512 for the

---

**Algorithm 2** Model-based Action-Gradient-Estimator TD3 (MAGE-TD3)

---

    **Input:** Initial buffer $\mathcal{B}$, parameters $\{\boldsymbol{\omega}, \phi_1, \phi_2, \boldsymbol{\theta}\}$, target parameters $\{\bar{\phi}_1 = \phi_1, \bar{\phi}_2 = \phi_2, \bar{\boldsymbol{\theta}} = \boldsymbol{\theta}\}$

  **for** each iteration **do**

    Collect transition $(s, a, s')$ acting according to exploratory policy $\pi_\epsilon(s) = \pi_{\boldsymbol{\theta}}(s) + \epsilon, \epsilon \sim \mathcal{N}(0, \sigma)$

    $\mathcal{B} \leftarrow \mathcal{B} \cup \{(s, a, s')\}$

    **for** each model learning step **do**

      $\boldsymbol{\omega} \leftarrow \boldsymbol{\omega} - \alpha_p \nabla_{\boldsymbol{\omega}} \ell(s, a, s'; \boldsymbol{\omega}), \qquad (s, a, s') \sim \mathcal{B}$

    **end for**

    **for** each policy optimization step **do**

      Extract state $s$ after sampling $(s, \cdot, \cdot) \sim \mathcal{B}$

      $\widehat{y} \leftarrow r(s, \pi_\epsilon(s)) + \gamma \min_{i=1,2} Q_{\bar{\phi}_i}(\widehat{s}, \pi_{\bar{\boldsymbol{\theta}}}(\widehat{s})), \qquad\qquad \widehat{s} \sim p_{\boldsymbol{\omega}}(\cdot|s, \pi_{\boldsymbol{\theta}}(s)), \epsilon \sim \mathrm{clip}(\mathcal{N}(0, \bar{\sigma}))$

      **for** $i \in \{1, 2\}$ **do**

        $\widehat{\delta}(s, a, \widehat{s}; \phi_i) \leftarrow \widehat{y} - Q_{\phi_i}(s, a), \qquad\qquad a = \pi_{\boldsymbol{\theta}}(s)$

        $\phi_i \leftarrow \phi - \alpha_Q \nabla_{\phi_i} \left( \left\| \nabla_a \widehat{\delta}(s, a, \widehat{s}; \phi_i) \big|_{a=\pi_{\boldsymbol{\theta}}(s)} \right\| + \lambda \left| \widehat{\delta}(s, a, \widehat{s}; \phi_i) \right| \right)$

        $\bar{\phi}_i \leftarrow \tau \phi_i + (1 - \tau)\bar{\phi}_i$

      **end for**

      **if** $t \bmod d = 0$ **then**

        $\boldsymbol{\theta} \leftarrow \boldsymbol{\theta} + \alpha_\pi \nabla_{\boldsymbol{\theta}} \min_{i=1,2} Q_{\phi_i}(s, \pi_{\boldsymbol{\theta}}(s))$

        $\bar{\boldsymbol{\theta}} \leftarrow \tau \boldsymbol{\theta} + (1 - \tau)\bar{\boldsymbol{\theta}}$

      **end if**

    **end for**

  **end for**

---

Pendulum and CartPole environment) states from the buffer of collected transitions, then sampling from the ensemble by first randomly selecting one of the members and then sampling an estimated difference between current and next state. The critic is trained by employing an Huber loss.

In MAGE-TD3, we employ the suggested hyperparameters of TD3: an action noise of $0.1$, a target noise of $0.2$, noise clipping to $0.5$ and a delay in the policy updates of $2$. During training, the actions that the actor executes in the environment are perturbed by Gaussian noise $\epsilon \sim \mathcal{N}(0, 0.1)$. We obtain the target networks for both actor and critic by Polyak averaging with decay $\tau = 0.995$.

For the reward estimation experiments, we employ a neural network with 3 hidden layers of 256 units (1 hidden layer with 128 units for the Pendulum and Cartpole environments) and swish activations We employ a discount factor of $\gamma = 0.99$.

For our experiment on the evaluation of gradients, we initially collect 200 transitions, then simply run the algorithms with standard settings but without any update of the actor. Every 10 steps, we collect 10 trajectories in the environment and average the error over them. We compute the ground-truth $\nabla_a G(s, a)|_{a=\pi(s)}$, with $G(s, a) = \sum_{t=0}^{H-1} \gamma^t r(s_t, a_t)|_{s=s_0, a=a_0}$ being the empirical return, by automatic differentiation, leveraging the differentiable oracle model. We then average, the resulting discounted error:

$$L(Q^\pi, \widehat{Q}) = \frac{1}{H} \sum_{t=0}^{H-1} \gamma^t \|\nabla_a G(s_t, a_t) - \nabla_a \widehat{Q}(s_t, a_t)\|_1. \tag{14}$$

We average this value across the 10 different trajectories.

Across all the experiments, despite the formulation we used throughout the paper, we employ a reward $r(s, a, s')$, which is thus also a function of the next state. Formally, it is possible to interpret the state-action reward we use throughout the paper as $r(s, a) = \mathbb{E}_{s' \sim p(\cdot|s, a)}[r(s, a, s')]$. For generating the performance plots, we evaluate, after every 1000 steps of environment interaction, the actor for 10 episodes and average the result. To improve presentation, we then uniformly smooth the resulting curves with a window size of 25.