[Reviews · NeurIPS 2020]

Review 1

Summary and Contributions: In "How to Learn a Useful Critic? Model-based Action-Gradient-Estimator Policy Optimization", a new objective for deterministic actor-critic methods is introduced in which the critic is optimized to minimize the norm of the action-gradient of the TD-error rather than on the TD-error directly. The new objective is theoretically justified and a practical implementation via a fitted dynamics model is empirically evaluated. Furthermore, the approach is applied to the more general case of unknown reward functions.

Strengths: The paper contributes a new perspective on off-policy actor-critic algorithms and is of clear interest to the NeurIPS community. The result that the difference between the true and approximated policy gradient is upper bounded by the action-value gradient of the policy evaluation error is very insightful. The theoretical grounding is solid.

Weaknesses: In order to alleviate the problem of erroneous model-predictions, the approach relies on a computationally expensive ensemble, since noise in the dynamics estimation is directly propagated to the value-function and the policy which can cause tremendous stability issues -- this is already already known for other combinations of value- and model-based methods. The rather uncommon hyperparameter setting (swish activation, huber loss for the critic, complicated training scheme of the model) and the limited evaluation on fairly easy control problems raises doubts whether the approach is easily applicable to more complex tasks. The unknown reward setting is furthermore only evaluated for the Pusher, the asymptotic performance only for the HalfCheetah-v2 environment. All other evaluations are capped after 1e5 transitions, which is especially bad for the Swimmer-v2 environment where the resulting return does not reflect any meaningful performance (to a degree that it should not be included as is in the camera ready). Additionally, it would be interesting to compare the introduced approach to n-step rollouts via model-based value estimation (see reference below) within TD3 and the same model-setting. Model-Based Value Estimation for Efficient Model-Free Reinforcement Learning. Feinberg et al., 2018.

Correctness: The proposed method is shown to be very effective on a variety of continuous control benchmarks. The approach is straightforward and should not be hard to reimplement, the exact hyperparameter setting is given in the appendix. However, the submission lacks a code submission.

Clarity: The paper is exceptionally well written and a pleasure to read. All derivations are easy to follow.

Relation to Prior Work: Due to its relation to Dyna, a few Dyna-based and model-based actor-critic methods are missing in the related work section [1,2,3]. Apart from that, the approach is very well put into context of current research and the differences are clear. [1] Continuous Deep Q-Learning with Model-based Acceleration. Gu et al., 2016. [2] Uncertainty-driven Imagination for Continuous Deep Reinforcement Learning. Kalweit and Boedecker, 2017. [3] Model-Augmented Actor-Critic: Backpropagating through Paths. Clavera, 2020.

Reproducibility: Yes

Additional Feedback: The broader impact section is very superficial and only partially discusses the societal implications of the work, however, the submission does not raise potential ethical concerns. ----- Post-rebuttal update: Thank you for the clarifications. However, despite the statement of the hyperparameter setting being common, it does seem to be rather approach specific. Thus, it is rather questionable whether any "reasonable setting" works well. In addition, the missing asymptotic evaluation (and the unkown results of such) and the points raised by fellow reviewers on TD-error vs. action-gradient are reasons for concern. This leads me to correct my scores downwards.


Review 2

Summary and Contributions: The paper proposes Model-based Action-Gradient-Estimator Policy Optimization (MAGE), a model-based policy optimization algorithm, which improves the performance of deterministic policy gradient by learning action-value gradient.

Strengths: The problem of improving deterministic policy gradient, learning better action-value gradient, and model based policy optimization are all important. This paper proposed a whole framework and the experiments look promising and improve over TD3 method.

Weaknesses: (1) The theory is not convicing. Proposition 3.1. is used to show that "the norm of the action-gradient of the policy evaluation error instead of its value that should be minimized". However, this is because the specific choice of the objective on the l.h.s. of this Proposition 3.1. Just saying "minimizing the TD-error does not guarantee that the critic will be effective at solving the control problem" is not convincing to me that this proposed objective is "a better objective function for critic learning". The author should show that minimizing the TD-error is not a good choice rather than just make plain claims. (2) The design of Eq. (7), together with the statement "minimization problem in Equation 6 is hard", looks to me is a contradiction with the claim made after Proposition 3.1. It looks to me lead to the conclusion that "TD error is important". And the authors do not have an explanation what is going on, and which one (TD error or gradient norm) really matters here. If just using TD or gradient norm did not work, then the conclusion should be both of them are important, rather than "the value gradient is a better objective than TD error". (3) The experiments look promising. However, first, it is for relatively simple tasks in Mujoco (not including difficult tasks like Humanoid). Second, the authors claim that "there is no intrinsic advantage in terms of sample-efficiency for model-based reinforcement learning" by using comparing TD3 and Dyna-TD3. This conclusion to me is just too hasty and not convincing. To make the (big) conclusion that "model-based RL has no advantage in terms of sample-efficiency than model-free RL", I think at least many more algorithms (of course just TD3 is far from enough) and other ways of learning and using models should be conducted and compared. Thus the subsequent claims of the performance is from the algorithmic design are not convincing to me. =====Update===== Thanks for the rebuttal. I agree with other reviewers that the Proposition 3.1 makes sense since critic learning is for providing a better policy gradient. Thus I would increase my score to 5. However, I still consider line135-138 and Appendix B.1, i.e., using action-gradient will fail, as an unclear point to me and somehow it makes the main point of this paper (using action-gradient should be a better choice for critic learning than TD-error) not that trustworthy. The authors said it is because of "local optima". And it is not clear why combining TD-error and action-gradient together will not suffer the same issue. Since the main point is to claim the importance of the action-gradient, this unclear point also seems an important issue.

Correctness: The claims are made not in a convincing way to me. It seems more investigations are needed to make things clear.

Clarity: The written and presentations are clear.

Relation to Prior Work: The related work discussion is thorough.

Reproducibility: Yes

Additional Feedback:


Review 3

Summary and Contributions: *** post rebuttal*** I maintain my score for weak acceptance. I do believe the estimator is not novel (the framework of "Credit assignment techniques in stochastic computation graphs" is the same as MAGE, i.e. providing valid and lower variance estimator of policy gradients, and the very same estimator is presented as an example in that paper. All papers I mentioned also cite Werbos, but that's not the point I was making). But the focused investigation of this particular estimator, along with the numerical experiments, and the theoretical result, justify the acceptance of the paper in my eyes. Two additional points I wanted to make: - the paper claims that learning a critic then computing its gradient is not as principled as directly learning the value gradient, since the policy gradient error will be dominated by the error in the value gradient. This is true, but the authors need to highlight and make it very clear this is not true for policy gradients in general, but for reparametrized (e.g. deterministic) policy gradient in particular. The error of a reinforce-like estimator would be driven by the TD error. While powerful in practice, it is not always true that DPG estimators have lower variance than 'regular' PG ones, so it would be best to make it very clear that the statements only apply because the authors consider a DPG-like scheme. - While it is true there is no particular reason to trust the gradients obtained by differentiating a critic which was learned by minimizing the critic error, it is also not clear why we should trust the gradients of the model, which will typically be learned by minimizing the model error. *** The paper studies the problem of learning a critic used for a DPG-style algorithm (differentiating through the critic to provide a gradient to the continuous action). The authors note that the traditional approach (regression of the critic against a valid target, then differentiating it) does not guarantee that the critic is useful, as it's the gradient of it we are interested in. They show the error in DPG is related to an error between between the true Q function gradient and the critic gradient. Since a target for the gradient of the Q function is not immediately available, they suggest building one by using a one-step model and a bootstrap argument.

Strengths: - The paper is clearly written and gives good background on an important problem in DPG-style algorithms. - The proposed approach is mostly sensible and is relatively simple to implement (in problems where state is well known). - Good experimental results on standard continuous controls benchmarks.

Weaknesses: - The novelty is somewhat limited: the challenge of learning the value gradient is mentioned at least in [1], and the technique to provide a valid target for the value gradient in [2] and more generally in [3] (theorem 6, specialized to the approach of interest on page 26, 'for instance a one-step gradient critic..'), the use of Sobolev norm is suggested in [4]. - The paper suggests replacing the evaluation error \delta (which is inaccessible unless using whole-horizon model rollouts) by the TD error \hat(\delta). While this is probably only a notation error, as the way it is presented in the paper, the loss appears invalid. In TD-learning for learning a Q function, the target can be written as r+Q(x',a'); however, when using it inside an L2 loss, it is important to not differentiate into the target (this is often implemented using a stop_gradient, i.e. (Q(x,a)-stop_grad(r(x,a)-gamma Q(x',a')))^2. The same exact issue happens when learning value gradient: the target dr/da + dx'/da dQ(x',a')/dx' is a valid target, but should not be differentiated into (note it depends on the optimization parameters through Q). [1] Compatible Value Gradients for Reinforcement Learning of Continuous Deep Policies, Balduzzi et. al [2] Value gradients, Fairbank [3] Credit assignment techniques in stochastic computation graphs, Weber et. al [4] Sobolev training, Czarnecki et. al

Correctness: Yes

Clarity: Yes

Relation to Prior Work: Yes

Reproducibility: Yes

Additional Feedback: The approach for building a valid target for the value gradient can be generalized to a k-step rollout (model for k-step, then bootstrap), or even a mix with a TD(lambda) like approach for value gradient. It would be interesting to see if deeper rollouts help out in some environments. Similary, the (1-step, or k-step) model-based target can directly be used instead of dQ/da, as suggested by [1], [2]. It is worth mentioning this is the case since there is a strong methodological connection between both (a valid bootstrap target can always be used to learn a critic/gradient critic or as replacement critic/gradient critic in a policy gradient scheme) [1]Imagined value gradients: Model-Based Policy Optimization, Byravan et. al [2] Model-augmented actor-critic: backpropagating through paths, Clavera et. al


Review 4

Summary and Contributions: In this work the authors propose a new actor-critic learning rule which they call Model-Based Actor Critic (MAGE). Specifically, they focus on improving upon the deterministic policy gradient methods in continuous control problems. While previous methods involved obtaining a critic using Temporal Difference learning and then updating the actor by computing the gradient of the learned critic, this work proposes a new approach which directly attempts to learn the critic gradient by employing a learned model. The authors show that such an approach is theoretically grounded and it leads to more data efficient learning.

Strengths: The authors back the claims of this new approach both from a theoretical and experimental standpoint. Their method is appealing, since explicitly optimizing for the quantities that matter usually leads to better performance. This work also spends enough time examining the different parts of the proposed method and tries to provide sufficient intuition for the experimental results. The paper is well structured and written.

Weaknesses: The one weakness of this work, was that the authors did not spend any time investigating the role of model accuracy in the performance of the learning rule. Obtaining an accurate environment model can be a really computationally expensive and challenging task in many interesting environments, and investigating how a poor model can affect the learning process would be a great addition to this work.

Correctness: Yes

Clarity: The paper is well written.

Relation to Prior Work: There is adequate discussion of previous contributions.

Reproducibility: Yes

Additional Feedback: It would be useful to provide quantitative results regarding the quality of the model. For example: How good are the samples produced by the model ? How does the quality of the model affect learning ? How does the method change when a deterministic model is used instead?

[Author Response · NeurIPS 2020]

We thank the reviewers for their comments and suggestions and for finding our theoretical considerations insightful and
solid [**R1**, **R3**, **R4**], our experimental results valuable [**R2**, **R3**, **R4**] and our paper well-written [**R1**, **R3**, **R4**].
**About multi-step rollouts**  We appreciate the suggestion of **R1** and **R3** on multi-step rollouts. We are curious as well
but we opted to leave it to future work: the goal of the paper is to show that model-based methods can be used to
directly learn the action-gradient of the value function, and we intentionally designed the simplest algorithm possible to
show this point. Using multi-step methods for computing either the gradient of the TD-target or the value gradient itself
is orthogonal to what we propose in MAGE, and would introduce additional complexity and hyperparameters (e.g.,
rollout length); we believe this would distract the reader from the main contribution of the paper. Nonetheless, we agree
that combinations of MAGE and these approaches can be particularly fruitful.
**R1:** ∗ The setup of MAGE follows common practices in (MB)RL: ensembles of models and the swish activation have
been used since PETS [7], and, more recently, in (SotA) MBPO [20], whose training scheme is very similar to ours;
the Huber loss has been shown to improve stability in DQN (2015). Those choices are known to be effective, but any
reasonable setting is enough for MAGE to perform well (see, e.g., the robustness of MAGE to the choice of $\lambda$ in Fig. 7
in Appendix B.3 and the additional experiment with different model architectures in Fig. R1). Therefore, we did not try
to cherry-pick the settings nor did any overly extensive hyperparameter search.
∗ Concerning the experiments on the unknown reward function setting, notice that Fig. 6 (in Appendix B.2) contains
results for all the tasks. We only included a single environment (Pusher-v2) in the main paper in order to save space.
∗ We indeed show the asymptotic performance on HalfCheetah-v2 only since our focus is on sample-efficiency;
nonetheless, we plan to include a larger number of steps (also for Swimmer-v2) in the final version of the paper.
∗ We will include the suggested references into the paper. See also **About multi-step rollouts**.
**R2:**  **(1)** The reviewer suggests that the paper should first "show that minimizing the TD-error is not
a good choice".    Notice, however, that despite being commonly used and thought of as "intuitive",
the minimization of the TD-error in DPGs is just a heuristic and lacks proper theoretical justification ever since its
first use in [42]. In contrast, our Proposition 3.1 inspires a reliable theoretically-grounded objective for learning a critic.
Furthermore, Fig. 3 shows indeed that minimizing the TD-error can lead to a critic being far away from the ideal one.
**(2)** We find no contradiction in our line of reasoning: we state that, despite the objective suggested by Proposition 3.1
is theoretically sound, there are practical issues that make the optimization problem hard. As written in the paper, we
interpret the TD-error as a regularization term with almost no additional overhead, but we do not exclude the existence
of other practical strategies for minimizing the bound in Proposition 3.1 without the use of the raw TD-error. We agree
that in MAGE the minimization of the TD-error is also important but we clearly showed with our experiments that the
*overall* objective is better than the TD-error alone.
**(3)** We did not write that "model-based RL has no advantage in terms of sample-efficiency than model-free RL". Instead,
we said that this advantage is not *intrinsic* and it is instead "highly environment and algorithm-dependent". We find
this statement uncontroversial in the light of other recent works in MBRL (see, e.g., [20,52]).
**R3:** ∗ We carefully positioned our approach in the literature and discussed similarities and differences w.r.t. most of
the works listed by R3. In particular, we directly cite Balduzzi et al. (2015) for proposing a solution to the problem of
value gradient learning. Notice, however, that MAGE provides a completely different and more direct way to address it.
Moreover, in our Related Works section, we acknowledge Werbos who pioneered (in 1977!) value gradient learning,
preceding Balduzzi et al's, Fairbank's and Weber et al's work by decades. Having said that, notice that these works lack
our theoretical motivation and belong to significantly different algorithmic frameworks. Likewise, Sobolev training
and related concepts in value gradients mainly share *technical tools* with MAGE, but they have different algorithmic
implications. Therefore, we are truly downhearted by the assertion regarding the alleged limited novelty of MAGE.
∗ Concerning the gradient stopping, we indeed use it: see Algorithm 1, in which target parameters $\bar{\phi}$ are employed
during the computation of $\widehat{\delta}$, a convenient notation for the gradient-stopping operation on the target $r(s, a) + \gamma Q_{\widehat{\phi}}(\widehat{s}, a')$
and its gradient w.r.t. the action $a$. Nonetheless, as suggested, we will make the gradient stopping more explicit.
∗ See **About multi-step rollouts**.
**R4:**  We agree with the reviewer that the quality of the learned model is of paramount importance for most MBRL
algorithms, whose performance, generally, deteriorates when the model is not enough expressive for a given task.
Thus, as suggested, we performed an additional experiment to investigate how the
performance of MAGE (compared to the Dyna-TD3 baseline) is affected by the use of
less powerful models. We evaluated two versions of the model with reduced capacity: (i)
only 2 members in the ensemble, 4 hidden layers and 256 units per layer (-small suffix);
(ii) no ensemble (a single model), 2 hidden layers, 256 units per layer (-smaller suffix).
Recall that the original setting involves a more powerful model with 8 members in
the ensemble, 4 hidden layers, 512 units per layer (no suffix). The results, reported in
Fig. R1, show that MAGE is robust to the presence of a misspecified model: while a
simpler but still quite capable model does no harm to MAGE, a significantly smaller
model has a reasonable impact on the obtained average return. We will report results
for all environments in the final version.

Figure R1: Pusher-v2 (5 runs, 95% c.i.).

[Meta-Review · NeurIPS 2020]

This paper proposes a modification to the way that the critic is typically trained in DPG, optimizing the critic to minimize a bound on the error of the value gradient rather than the error of the value, with the targets computed using a learned model. R1, R3, and R4 highlight that the approach is interesting and well-motivated, with R1 praising the insightfulness of the approach. Although R3 questioned the novelty of the method, and R2 was unconvinced by some of the theoretical claims, I think that the authors have done a good job of demonstrating empirically that their approach improves upon the baseline. Additionally, even if the idea is not completely new, there is value in fleshing out an idea and getting it to work in practice. I believe this paper will be of broad interest to both model-free and model-based researchers in the RL community at and thus recommend acceptance. However, I ask the authors for the camera-ready to please make address the following points from R2 and R3: (1) be more explicit that Proposition 3.1 only applies to DPG and not to the general case of PG; (2) discuss the effect of bad gradients in the learned model; and (3) temper the claims about optimizing the action-gradient being the right thing to do, in light of the practical difficulties encountered when optimizing this (requiring the TD error to be included as a regularization term).